# A Novel Analytical Approach to Assessing Sorption of Trace Organic Compounds into Micro- and Nanoplastic Particles

**DOI:** 10.3390/biom12070953

**Published:** 2022-07-06

**Authors:** Julia Reichel, Johanna Graßmann, Oliver Knoop, Thomas Letzel, Jörg E. Drewes

**Affiliations:** 1Chair of Urban Water Systems Engineering, Technical University of Munich, 85748 Garching, Germany; julia.reichel@tum.de (J.R.); j.grassmann@tum.de (J.G.); oliver.knoop@tum.de (O.K.); t.letzel@tum.de (T.L.); 2Analytisches Forschungsinstitut für Non-Target Screening GmbH (AFIN-TS GmbH), Am Mittleren Moos 48, 86167 Augsburg, Germany

**Keywords:** microplastic, nanoplastic, desorption, sorption processes, TD-Pyr-GC/MS

## Abstract

Assessing the sorption of trace organic compounds (TOrCs) into micro- and nanoplastic particles has traditionally been performed using an aqueous phase analysis or solvent extractions from the particle. Using thermal extraction/desorption–gas chromatography/mass spectrometry (TD-Pyr-GC/MS) offers a possibility to analyze the TOrCs directly from the particle without a long sample preparation. In this study, a combination of two analytical methods is demonstrated. First, the aqueous phase is quantified for TOrC concentrations using Gerstel Twister^®^ and TD-GC/MS. Subsequently, the TOrCs on the particles are analyzed. Different polymer types and sizes (polymethyl methacrylate (PMMA), 48 µm; polyethylene (PE), 48 µm; polystyrene (PS), 41 µm; and PS, 78 nm) were analyzed for three selected TOrCs (phenanthrene, triclosan, and α-cypermethrin). The results revealed that, over a period of 48 h, the highest and fastest sorption occurred for PS 78 nm particles. This was confirmed with a theoretical calculation of the particle surface area. It was also shown for the first time that direct quantification of TOrCs from PS 78 nm nanoparticles is possible. Furthermore, in a mixed solute solution, the three selected TOrCs were sorbed onto the particles simultaneously.

## 1. Introduction

Due to the wide range of beneficial properties of polymers (such as low cost, durability, and light weight), it is hard to imagine our everyday life without plastics [1,2]. However, improper disposal of plastic products worldwide is causing more and more plastic waste to enter the environment [3]. Larger plastic litter is subsequently being converted into microplastics by the fragmentation of larger plastic pieces through photolytic, mechanical, and biological degradation processes, while the chemical structure remains intact [4,5,6]. Microplastic particles can further disintegrate into nanoplastic [7,8,9]. Recently, micro- and nanoplastics have been defined by an ISO standard (ISO/TR 21960:2020). According to this standard, microplastics are water-insoluble particles with a size of 1 µm to 1 mm and nanoplastics are particles smaller than 1 µm.

The porous polymer structure and high surface area of micro- and nanoplastics enables both the adsorption of chemicals onto the particle surface and the absorption into the particles [10]. Due to the hydrophobicity of plastics and their high surface-to-volume ratio, trace organic compounds (TOrCs) might sorb readily to micro- and nanoplastics. This can lead to the accumulation of TOrCs on plastic particles from the aqueous phase [2,9,11,12]. Thus, micro- and nanoplastics might act as vectors distributing persistent TOrCs into the environment and causing bioaccumulation in organisms [9,11,13,14]. A number of persistent organic pollutants have been found to readily sorb into polymer particles, such as polychlorinated bisphenyls (PCBs), dichlorodiphenyldichloroethylene (DDE), dichlorodiphenyltrichloroethane (DDT), and polycyclic aromatic hydrocarbons (PAHs) [11,15,16]. However, the environmental risk associated with exposure of microplastics carrying TOrCs also depends on their relative source contribution. One review reports that the amount of bioaccumulated hydrophobic organic chemicals from natural prey in rivers is higher than the amount of ingested microplastics [17].

The identification or quantification of polymers can be achieved either spectroscopically by Fourier transform infrared spectroscopy (FTIR) or Raman spectroscopy in terms of particle numbers and sizes, or by thermal analysis in terms of polymer masses [18,19,20,21]. Thermal analysis-based methods include, for instance, thermal desorption–gas chromatography mass spectrometry (TD-GC/MS) or pyrolysis–gas chromatography mass spectrometry (Pyr-GC/MS) [20,22,23,24]. However, here, only the polymer is being analyzed and not the sorbed TOrCs. To quantify sorbed TOrCs, two different approaches are commonly applied: either the sorbed TOrCs are analyzed indirectly by examining the aqueous phase under controlled conditions or the sorbed substances are extracted from the particles in specific extraction steps with solvents and subsequently analyzed [25,26,27,28,29,30,31,32]. A limitation of the extraction of TOrCs from particles with a subsequent analysis is that not all solvents are suitable for all polymers. For instance, methanol seems not to be suitable for the extraction of DDE from polypropylene (PP) particles and others [15]. A comprehensive summary of current analytical approaches was summarized in a recent review [33].

TD-Pyr-GC/MS offers the possibility to identify the sorbed TOrCs and polymers using one analytical setup [34]. For this purpose, the TOrCs are initially thermo-desorbed from the particles and analyzed by GC/MS. In the subsequent pyrolysis step, the polymer is pyrolyzed and can then be identified by its characteristic monomers. Due to a simple modification of the TD-Pyr-GC/MS not only the particles but also the aqueous phase concentration of TOrCs can be investigated by stir bar analysis (Gerstel Twister^®^) and TD-GC/MS.

The concept of a combined TD-GC/MS and TD-Pyr-GC/MS analysis is applied in this study for the first time. The degree of sorption of selected TOrCs, phenanthrene, triclosan, and α-cypermethrin on polyethylene (PE), polystyrene (PS), and polymethyl methacrylate (PMMA) is investigated with TD-Pyr-GC/MS. In all experiments, both the aqueous phase (TD-GC/MS) and the particles (TD-Pyr-GC/MS) are analyzed directly. The aim of this study is to investigate the degree of sorption of each TOrCs after 1 h, 24 h, and 48 h on representative micro- and nanoparticles.

## 2. Analytical Systems, Materials, and Methods

### 2.1. Instrumental Systems

For the analysis of aqueous solutions, stir bars (Gerstel Twister^®^) were employed and purchased from Gerstel GmbH & Co. KG (Mühlheim an der Ruhr, Germany). The TD-Pyr-GC/MS analysis was equipped with a Gerstel Thermal Desorption Unit (TDU) 2, a Gerstel Multi-PurposeSampler (MPS) robotic^pro^, a Cooled Injection System (CIS) 4 with C506, and an Agilent 7890B gas chromatograph equipped with an DB-5MS Ultra Inert column coupled to an Agilent 5977B MSD mass spectrometer.

The particles were analyzed using a TD-Pyr-GC/MS system. The setup was basically the same, except that a pyrolysis module was integrated into the TDU 2 [34]. In the thermodesorption phase (TD-GC/MS), the sample was first heated up to a temperature of 200 °C to desorb the TOrC. At −50 °C, the volatiles were trapped in the CIS and transferred to the GC column while heating to 200 °C. The GC method was adopted from Ochiai et al. (2005) [35]. The MS analysis was conducted in SIM mode. The same sample was now pyrolyzed at 800 °C (Pyr-GC/MS). Since not all volatile TOrCs completely desorb from the particle at a TD temperature, subsequent pyrolysis is essential. The pyrolysis products were trapped and released at 350 °C. The sample injection of the CIS was operated in split mode (100:1) to avoid contamination by polymer products. The sample was subsequently analyzed by GC/MS. The initial temperature was set to 50 °C and maintained for 2 min. The GC oven was subsequently heated to 320 °C with a gradient of 10 °C/min. This temperature was held for 3 min. The final evaluations were carried out using the TD-Chromatogram for TOrC analysis and the Pyrogram for polymer analysis. A more detailed method description can be found in Reichel et al., 2020 [34].

### 2.2. Materials

The reference micro- and nanoplastic particles were provided by BS-Partikel GmbH (Mainz, Germany). The spherical polystyrene particles in sizes of 78 nm and 41 µm, respectively, were supplied suspended in EtOH. The PE and PMMA particles with a size of 48 µm were available in a dry state. α-cypermethrin (CAS: 67375-30-8) was purchased from greyhoundchrom (Birkenhead, UK), phenanthrene (CAS: 85-01-8), and triclosan (CAS: 3380-34-5) from Sigma Aldrich (Taufkirchen, Germany). The deuterated standards phenanthrene-d10 (CAS: 1217-22-2) and cypermethrin-(phenoxy-d5) were purchased from Sigma Aldrich (Taufkirchen, Germany). Triclosan-d3 (CAS: 1020719-98-5) was purchased from Toronto Research Chemicals (Toronto, Canada). All chemicals were dissolved in methanol and stored at 4 °C. Methanol (≥99.8%) in HPLC-grade was purchased from VWR (Ismaning, Germany). For the simulation of real environmental conditions, tap water from Garching (Germany) was used for all sorption experiments. For the weighing of the samples, a Sartorius Cubis^®^ Ultramicro Balance (Göttingen, Germany) was used. For filtration, nucleopore hydrophilic membrane filters (0.03 µm pore size) were purchased from Whatman/GE Healthcare (Marlborough, MA, USA).

### 2.3. Sample Preparation for Sorption Processes Experiments

In an aqueous solution, the selected TOrCs (phenanthrene, triclosan, and α-cypermethrin) were incubated with the defined micro- or nanoplastic particles. Depending on the experiment, either the incubation time or the TOrC concentration were varied. In addition, an experiment was performed with a mixture of the three selected TOrCs (phenanthrene, triclosan, and α-cypermethrin). The three model substances were selected based on their ecotoxicological relevance [36,37,38,39]. The particle concentration was always 1 g/L for all polymers and sizes. After incubation, the suspensions were filtered to separate the aqueous phase from the particulate phase. Both the aqueous and particulate phases were analyzed to establish a mass balance between sorbed and unsorbed TOrCs.

The entire sample preparation workflow is shown in Figure 1.

The respective methods are explained in the following sections.

#### 2.3.1. Analysis of the Aqueous Phase

After incubation and filtration, the filtrate was diluted 1:10 for quantitative analysis. The deuterated standards phenanthrene-d10 (50 ng/L), cypermethrin-(phenoxy-d5) (0.1 mg/L), and triclosan-d3 (0.01 mg/L) were added corresponding to the TOrC filtrate to obtain a final volume of 10 mL. A stir bar sorptive extraction (SBSE; Gerstel Twister^®^, Mühlheim an der Ruhr, Germany) was used for the TD-GC/MS quantification of the TOrCs. The Gerstel Twisters^®^ were coated with a polydimethylsiloxane (PDMS, film thickness 0.5 mm, length 10 mm) layer. In all experiments, the Gerstel Twisters^®^ were stirred on a Thermo Fisher (Waltham, MA, USA) magnetic stirrer (15 positions) for 1 h at room temperature and 1000 rpm. The Gerstel Twister^®^ was removed, washed with ultra-pure water, and dried with a lint-free tissue. The Gerstel Twister^®^ was then transferred to the thermal desorption tube and analyzed by TD-GC/MS.

Calibration curves were prepared in each case using the appropriate deuterated standards to quantitatively determine the TOrCs in the filtrate. In order to eliminate the influence of filtration, the calibration standards were also filtered.

#### 2.3.2. Particle Analysis

After filtration, the particles were scraped off the filter with a spatula, freeze-dried for 20 min, and stored at 4 °C for a maximum of 24 h. These were then weighed directly into the pyrolysis tubes, with a maximum of 80 µg to avoid a system overload. This sample was then analyzed by TD-Pyr-GC/MS.

#### 2.3.3. Sorption Processes as a Function of Time

For the determination of the sorption of the selected TOrCs as a function of time (1 h, 24 h, and 48 h), 10 mg of particles were suspended in 10 mL of tap water in each case. The TOrCs phenanthrene, triclosan, and α-cypermethrin were added with a final concentration of 1 mg/L. The stock solution of the TOrCs was previously prepared in methanol. The suspension was shaken for 1 h, 24 h, and 48 h at room temperature at 1000 rpm. All experiments were performed in quadruplicates.

#### 2.3.4. Sorption Processes with Different TOrC Concentrations on Nanoparticles

In order to determine whether TD-Pyr-GC/MS can also be used to reliably determine concentration differences on the particles, three concentrations (1 mg/L, 5 mg/L, and 10 mg/L) of each of the selected TOrCs (phenanthrene/triclosan/α-cypermethrin) were sorbed onto the PS 78 nm particles in aqueous solution (10 g/L), respectively. Since rapid sorption into the PS 78 nm particles was observed within 1 h in previous experiments, the incubation time was set to 1 h. Both the aqueous phase (TD-GC/MS) and the particles (TD-Pyr-GC/MS) were analyzed.

#### 2.3.5. Sorption Processes with Mixtures of TOrCs

To investigate how the three TOrCs phenanthrene, triclosan, and α-cypermethrin affect each other, the three TOrCs were added simultaneously to the PS 78 nm and PE 48 µm particles (10 g/L) at concentrations of 1 mg/L and 10 mg/L, respectively. After an incubation period of 1 h, the samples were filtered and analyzed by TD-GC/MS and TD-Pyr-GC/MS.

### 2.4. Evaluation of the TD-GC/MS and TD-Pyr-GC/MS Data

*TD-GC/MS:* The mass spectrometer operated in SIM mode and the TOrCs phenanthrene (*m*/*z* 178), triclosan (*m*/*z* 290), and α-cypermethrin (*m*/*z* 163), and their corresponding deuterated standards were analyzed (Table 1).

*TD-Pyr-GC/MS:* In the thermodesorption step, analysis was performed in a combined SIM/full scan mode to identify the selected TorCs in the SIM mode and, at the same time, also to identify any characteristic substances of the polymers in the full scan mode. Pyrolysis was performed in full scan mode to identify potential carryover or contamination of polymer products. Moreover, pyrolysis can also identify the TOrCs (e.g., phenanthrene) that do not completely desorb in the TD.

Data analysis was performed using Mass Hunter Workstation software (Ver.B.08.000, Agilent) for TD-GC/MS and TD-Pyr-GC/MS analysis. Compound identification was validated via MS spectra, and a NIST database comparison was performed.

Specific mass spectrometric signals were selected for the identification of the selected TOrCs in the aqueous and in the particulate phases via TD-GC/MS (Table 1). A more detailed description of the data analysis can be found in Reichel et al., 2020 [34]. The data were standardized (peak area over weighed particle mass in the pyrolysis tube) to provide a basis of comparison for the sorption of TOrCs on polymers.

## 3. Results and Discussion

The aim of the study was to establish the analysis of TOrCs directly from particles and to validate it with an aqueous phase analysis. For the sorption tests with phenanthrene, triclosan, and α-cypermethrin, a TOrC concentration of 1 mg/L was applied in each case. PE 48 µm, PMMA 48 µm, PS 41 µm, and PS 78 nm were used as reference particles. The concentration of the particle suspension was 1 g/L.

### 3.1. Sorption Behavior of Phenanthrene, Triclosan, and α-Cypermethrin onto Reference Particles

Up until now, most sorption studies have investigated either only the aqueous or the particulate phase [25,26,27,29,31,40]. So far, no mass balances of the sorbed substances, consisting of aqueous and particulate phases, have been performed. In the following, sorption of the TOrCs phenanthrene, α-cypermethrin, and triclosan in the aqueous phase and onto the particles after incubation times of 1 h, 24 h, and 48 h are considered.

#### 3.1.1. Phenanthrene

Phenanthrene is a non-polar chemical and easily sorbs into non-polar polymers. Consequently, the results shown in Figure 2a confirm that phenanthrene is no longer present in the aqueous filtrate of the PE 48 µm, PS 78 nm, and PMMA 48 µm particles already after an exposure time of 1 h. For PS 41 µm particles, the phenanthrene concentration in the aqueous filtrate decreases over time. Considering the sorption onto the different polymers illustrated in Figure 2, it can be seen that, despite large deviations, the sorption onto PE 48 µm (Figure 2b) and PS 78 nm (Figure 2a) particles is already completed after 1 h. Deviations can occur due to the weighing of the particles or due to incomplete pyrolysis of the particles. The sorption results of the particle phase analysis (TD-Pyr-GC/MS) are supported by the aqueous phase results (TD-GC/MS). A slight increase in phenanthrene concentration on the PS 41 µm particles from 1 h to 48 h is indicated.

Compared with PS and PE, PMMA is relatively polar. Therefore, even after 48 h, hardly any sorption of phenanthrene onto the particles was noticed (Figure 2c). This is confirmed by the remaining high concentrations of TOrCs in the aqueous phase.

Compared with the other polymers studied, PE is the least polar. Therefore, its high affinity for phenanthrene is to be expected. In the case of the polymer PS and phenanthrene, which are both aromatic compounds, a non-covalent interaction occurs by “stacking” benzene rings and π–π interactions. Phenanthrene, with a rigid planar surface, may approach the PS particle surface [41,42]. In addition, there is an additional effect of benzene rings on sorption capability, with absorption being enhanced by a greater distance between polymer chains due to the benzene rings [43]. Comparing the similar sized particles of PE 48 µm and PS 41 µm, the sorption onto the PE particles is clearly increased. According to Pascall et al. (2005), this might be due to the different distances between the polymer chains in PS and PE [43]. The polymer backbone of PS consists of a benzene molecule, while that of PE consists of hydrogen atoms. As a result, the segmental mobility within the polystyrene chains is temperature dependent, whereas that of PE is not. If the segment mobility is high and the distance between the polymer chains is large, the TOrCs can easily diffuse into the polymer matrix. In addition, the segmental mobility of PS is reduced by the presence of benzene.

Considering the sorption of phenanthrene based on the aqueous and particulate phase data over a period of 48 h (sampling: 1 h, 24 h, and 48 h), the sorption follows the following order: PMMA 48 µm < PS 41 µm < PE 48 µm < PS 78 nm.

#### 3.1.2. Triclosan

Due to its fairly high symmetry, triclosan shows little polarity (Table 1). After only 1 h, almost all of the triclosan is sorbed onto the PS 78 nm nanoparticles (Figure 3a). The aqueous phase and the particle data have been examined. In contrast to the PS nanoparticles, sorption onto the PS microparticles (PS 41 µm) is not yet complete even after 48 h (Figure 3d). The data of triclosan and the comparably polar PMMA (Figure 3c) clearly indicate that sorption onto the particles has not yet occurred after 48 h.

Due to the benzene rings in the structure of triclosan, it can form π–π interactions with those of PS [26]. Additionally, the higher sorption of triclosan onto the nanoparticles (PS 78 nm) compared with the microparticles (PS 41 µm) is clearly evident. This confirms the results of Li et al. (2019), who reported an increased sorption capacity of triclosan with decreasing PS particle size [44]. According to the literature, the sorption of triclosan onto the polymers occurs mainly due to hydrophobic, hydrogen-bonding, and π–π-bonding interactions [45,46].

The analysis of the aqueous phase concentration of triclosan revealed the following order: PMMA 48 µm < PS 41 µm < PE 48 µm < PS78 nm. If only the particle phase is considered, the following order appears: PMMA 48 µm = PS 41 µm < PE 48 µm < PS78 nm. Comparing the same polymers PS, a significantly higher sorption onto the nanoparticles is evident. These results are supported by Ma et al. (2019), reporting that particle size affects the sorption behavior of triclosan [47], confirming that smaller particles provide a larger surface area for sorption.

#### 3.1.3. α-Cypermethrin

α-cypermethrin is a mixture of the two enantiomers 1*R*-*cis*-*αS* and 1*S*-*cis*-α*R*. As reported by Qin and Gan/2007), the substances can undergo isomerization in some organic solvents such as methanol [48]. Since the standard solutions of α-cypermethrin were prepared in methanol, it is likely that they also contain the diastereomers. The observed signals in the chromatogram are therefore caused by the isomers, where the two sets of enantiomers each produced one signal in the chromatogram. The same isomerization has also been reported due to the high temperatures in the GC inlet, which could also cause the presence of two signals in the chromatogram [49]. This could also affect the final evaluation of α-cypermethrin, especially in the aqueous phase with the deuterated standard.

The concentration of α-cypermethrin in the aqueous phase is consistently very low for all polymers (Figure 4a–d). Since the deuterated standard cypermethrin-(phenoxy-d5) was used as a reference in the aqueous phase, this could have influenced the measurements due to the presence of four isomers. Based on the polymer data, the highest sorption occurs on the PS 78 nm particles (Figure 4a). However, deviations over 1 h, 24 h, and 48 h are very large. For the PE 48 µm (Figure 4b), PMMA 48 µm (Figure 4c), and PS 41 µm (Figure 4 d) particles, the sorption onto the particles is lower.

Compared with phenanthrene, the benzene rings of the α-cypermethrin cannot bind as effectively to the hydrogen groups of the PE due to additional structural elements. Furthermore, α-cypermethrin does not have a planar structure, which hinders the binding of the benzene rings to PS. Due to the amorphous structure of PMMA, α-cypermethrin could absorb more into PMMA particles.

Considering the degree of sorption on the particles, the sorption after 48 h on all polymers is approximately the same (Figure 4a–d). However, the PS 78 nm nanoparticles show large deviations. Within the two measuring points at 1 h and 24 h, the picture is different. The sorption to the PS particles is highest for PE 48 µm = PMMA 48 µm < PS 41 µm < PS 78 nm.

### 3.2. Sorption of TOrCs as a Function of the Calculated Particle Surface Area

The surface area property describes the outer surface area of the particle that is available for adsorption of TOrCs (see Section 3.1). The reference PS particles (PS 78 nm and PS 41 µm) are spherical; the PE 48 µm and PMMA 48 µm particles are not. For an estimation of the adsorption of the TOrCs on the particle surface, a spherical shape of all particles is nevertheless assumed for this calculation. The calculation for the particle surface is performed with the applied amount of 10 mg and the general formula for the calculation of a spherical surface. The calculation of the TOrC surface is considering the calculated van der Waals surface of the respective TOrCs for an amount of 10 µg (corresponds to the concentration of 1 mg/L). Based on both surfaces, the ratio between TOrCs and particle surface is derived (see Table 2).

#### 3.2.1. Phenanthrene

The rapid sorption of phenanthrene onto the PS 78 nm nanoparticles can be explained by the low occupancy of the surface (ratio phenanthrene/PS 78 nm: 0.12). Considering the microplastic particles PMMA 48 µm and PS 41 µm, the ratio of TOrC occupancy by particle surface is much higher, which explains the slower sorption. The PE 48 µm particle exhibited just after 1 h approximately the same peak area/weight as the PS 78 nm particles, although the ratio is 540 times higher. However, the PE 48 µm particles are not spherical, so the reported ratio is only an estimate.

#### 3.2.2. Triclosan

Similar to phenanthrene, triclosan occupancy is lowest (ratio of triclosan/PS 78 nm: 0.07) on the PS 78 nm particles. Therefore, sorption onto the particles is also completed after 1 h. The sorption on PMMA 48 µm and PS 41 µm particles is the same in both cases, which can also be explained by the TOrC/particle ratio. The sorption onto the PE 48 µm particles is greater despite the higher TOrC/particle ratio, but again, the non-spherical nature of this reference particle must be taken into account.

#### 3.2.3. α-Cypermethrin

Based on the particle data (Figure 4a), it can be clearly seen that, here, too, the highest degree of sorption occurs on the PS 78 nm particles. This is explained by the low surface coverage with α-cypermethrin (ratio α-cypermethrin/PS 78 nm: 0.05). The ratio of triclosan to PE 48 µm, PS 41 µm, and PMMA 48 µm is approximately the same. This is confirmed by the sorption results of the particle data.

These results are confirmed by other studies reporting that the sorption of TOrCs on nanoplastics is much stronger than on microplastics [26,50]. Sorption of polychlorinated biphenyls (PCBs) to nano-PS (70 nm) was 1–2 orders of magnitude stronger than to micro-PE (10–180 µm) due to the higher aromaticity and larger surface-to-volume ratio of nano-PS [26]. In the further study, three different synthetic musks and their sorption onto polypropylene (PP) particles of different sizes (2–5, 0.85–2, 0.425–0.85, or 0.125–0.45 mm) were investigated. Again, the adsorption capacity was found to increase with smaller particle size [50].

### 3.3. Sorption of Selected TOrCs as a Function of TOrC Concentrations on Nanoparticles

Based on the previous experiments, the PS 78 nm particles were selected to conduct further experiments with different concentrations (1 mg/L, 5 mg/L, and 10 mg/L) of TOrCs to be sorbed onto the PS 78 nm particles. The aim of the experiment was to evaluate whether a quantitative analysis of the sorbed TOrCs on the nanoparticles is in principle feasible by TD-Pyr-GC/MS.

#### 3.3.1. Phenanthrene + PS 78 nm Particles

The phenanthrene concentration in the aqueous phase and on the particles was investigated (Figure 5a). For the initial concentration of 1 mg/L, there is no more phenanthrene in the aqueous phase after an incubation time of 1 h. Concentrations of less than 0.2 mg/L were observed for the 5 mg/L and 10 mg/L concentrations after 1 h of exposure in the aqueous phase. In the TD-Pyr-GC/MS particle analysis, a distinct increase in sorbed phenanthrene was noted. A quantification of this TOrC should therefore be possible.

#### 3.3.2. Triclosan + PS 78 nm Particles

The triclosan concentration in the aqueous solution and the corresponding amount on the particles of the initial concentrations of 1 mg/L, 5 mg/L, and 10 mg/L were studied (Figure 5b). For all initial concentrations, the final concentration after 1 h incubation is below 1 mg/L. Considering the particle analysis (TD-Pyr-GC/MS), also here, a clear increase in the triclosan concentration on the particles can be seen, so that quantification of triclosan on the particles is possible.

#### 3.3.3. α-Cypermethrin + PS 78 nm Particles

The concentration in the aqueous phase and in the particulate phase with the TOrC α-cypermethrin after 1 h incubation with PS 78 nm particles and initial concentrations of 1 mg/L, 5 mg/L, and 10 mg/L were investigated (Figure 5c). Compared with phenanthrene and triclosan, the remaining concentrations and the deviations in the aqueous phase are significantly higher. Based on these results (Figure 5c), a quantification of the sorbed amount on the particles is possible.

### 3.4. Sorption of the TOrC Mixture onto Reference Particles

Most studies on sorbed TOrCs on micro- or nanoplastic particles examine single substances [10,30,44,51,52,53]. However, it is not to be expected that TOrCs will occur individually in the aquatic environment, but that they are always present in mixtures [16,26]. If a mixture of substances is used, the sorption capacity of the single substance may be affected. Therefore, in an additional experiment, the three selected TOrCs phenanthrene, triclosan, and α-cypermethrin were sorbed simultaneously at concentrations of 1 mg/L and 10 mg/L onto the polymers PS 78 nm and PE 48 µm (1 g/L each). These particles were selected because the sorption equilibrium had been rapidly established after 1 h in the sorption experiments. In each of the experiments, the aqueous phase (TD-GC/MS) and the particles (TD-Pyr-GC/MS) were analyzed.

The three TOrCS were sorbed onto PS 78 nm particles with an initial concentration of 10 mg/L (Figure 6a). The concentrations of the individual TOrCs (α-cypermethrin phenanthrene and triclosan) are compared with the mixed sorbed substances. Considering the results from the aqueous phase and from the particle measurements, only minor differences of the mixed measured TOrCs compared with the single substances were observed which are in the range of the measurement fluctuations. In the case of these three substances, the sorption capacity of the selected TOrCs neither increased nor decreased. The calculated particle surface areas (Section 3.2) indicate that the surface of the nanoparticles is only slightly occupied, and therefore, no agonistic or antagonistic effects occur.

Analogous to the sorption on PS 78 nm particles, the single and mixed substances were sorbed onto PE 48 µm particles (Figure 6b). Based on the surface occupancy, a final concentration of 1 mg/L was selected for all TOrCs. The sorption of α-cypermethrin is not affected by the presence of the other TOrCs. Both in the aqueous phase and on the particles, the sorption differs only marginally. Phenanthrene sorbs more strongly as a single substance on PE 48 µm. This could indicate an antagonistic effect, which was already observed by Bakir et al. (2012) for phenanthrene and DDT [54]. Triclosan, similar to α-cypermethrin, is hardly affected by the presence of the other TOrCs. Both the aqueous phase and the particle analysis indicate similar sorption.

Comparison of the sorption of the mixing experiments of PS 78 nm and PS 48 µm particles with an initial TOrC concentration of 10 mg/L shows that the degree of sorption of all TOrCs is higher on the PS nanoparticles (Figure 6c). This is reflected in the aqueous phase as well as in the particle phase.

### 3.5. Stability and Reproducibility of Data in TD-GC/MS and TD-Pyr-GC/MS Analysis

The quantitative analysis of an aqueous sample with stir bar sorptive extraction (via Gerstel Twister^®^) and a GC/MS analysis is an established routine procedure [38,55,56]. Nevertheless, high standard deviations occurred in some cases. One reason for this could be the filtration step. Although nanofilters were used, it cannot be ruled out that smaller particle fractions enter the aqueous phase and interact with the Gerstel Twisters^®^. Additionally, the duration of filtration varies and depends on how quickly the filter clogs.

Regarding the TD-Pyr-GC/MS data of the sorbed TOrCs, they show high standard deviations with partly low reproducibility, especially in the incubation experiment over 48 h. However, only a concentration of 1 mg/L was investigated in these experiments. If the quantification experiments with concentrations of 1 mg/L, 5 mg/L, and 10 mg/L are considered, the data still have large deviations in some cases, but the different sorbed concentrations are clearly visible. In these experiments, it was demonstrated for the first time that quantification of TOrCs directly from the micro- and nanoparticle is possible. Previous analyses directly from the microplastic particle have so far only been able to identify substances such as polymer additives [22,23] and to quantify phthalates [57]. Quantification of TOrCs on nanoparticles has not been possible up until now.

## 4. Conclusions and Outlook

Based on the results of this study, the combination of TD-GC/MS and TD-Pyr-GC/MS can be applied as a novel and quick method for the analysis and quantification of sorbed TOrCs on micro- and nanoparticles and for polymer identification [34]. With the analysis of both phases, the concentration in the aqueous phase and the peak area/weight on polymers could be shown for the first time.

Sorption of the three selected TOrCs (phenanthrene, triclosan, and α-cypermethrin) onto three different polymer types and sizes (PS 78 nm, PS 41 µm, PE 48 µm, and PMMA 48 µm) within 48 h was demonstrated. The results showing that the highest sorption occurs on the PS 78 nm nanoparticles is supported by the calculation of the high surface-to-volume ratio of the particles.

It was demonstrated for the first time that quantification directly from the particles is possible using TD-Pyr-GC/MS. By showing clear peak area/weight differences for all three selected TOrCs at concentrations of 1 mg/L, 5 mg/L, and 10 mg/L, a calibration based on TOrC-loaded nanoparticles can be constructed. All data were statistically analyzed for significant outliers using the Dixon Q test and the Grubbs test at a level of 0.05. The quantification of TOrCs on particles could be particularly interesting for use in ecotoxicological assays or in experiments using controlled laboratory-scale wastewater treatment plants.

By means of TD-Pyr-GC/MS, quantitative experiments especially regarding environmental samples, could be performed in the future. Nevertheless, sample preparation must be taken into account here and will need a specific design for each analytical question. The samples should be free of organic material, such as biofilms. At the same time, the presence of potentially interfering inorganics should be avoided. Additionally, the investigation of the amount truly sorbed into the particles under the influence of various factors such as pH, temperature, or salinity should be feasible using this analytical method. For future experiments, it would be useful to consider different polymer types in the nanometer range. As long as no chemical interactions during desorption occur, the sorbed TOrCs can be quantified directly from the particle regardless of the polymer type and extraction methods.

In the future, the experimentally obtained sorption data obtained by TD-Pyr-GC/MS analysis could still be validated by modeling (e.g., pp-LSER). However, so far, these modeling approaches for sorption experiments have been performed mainly with microplastic particles and not with nanoplastic particles, so the number of databases is likely to be limited [58,59].

## Figures and Tables

**Figure 1 biomolecules-12-00953-f001:**
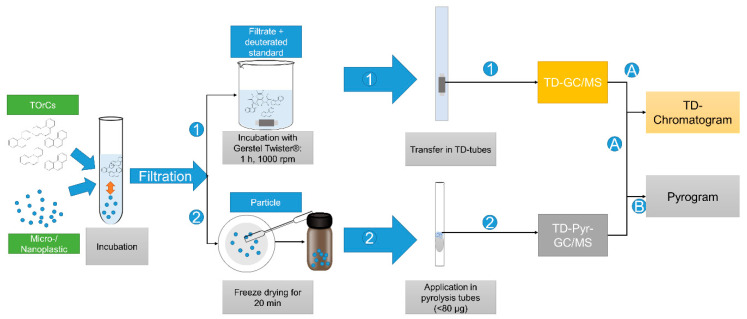
First, the TOrCs and microplastic particles are incubated in aqueous solution. Then, the sample is filtered. The filtrate (1) is mixed with the deuterated standard and stirred for 1 h with the Twister. The Twister is added to the TD tube and analyzed via TD-GC/MS. The TOrCs are analyzed via the TD chromatogram (**A**). The particles (2) are scraped off the filter with a spatula, placed in a vial, and freeze dried. The dried particles are weighed directly into the pyrolysis tube and analyzed by TD-Pyr-GC/MS. The TD chromatogram is used to evaluate the volatiles (**A**), and the pyrogram is used for the polymers (**B**).

**Figure 2 biomolecules-12-00953-f002:**
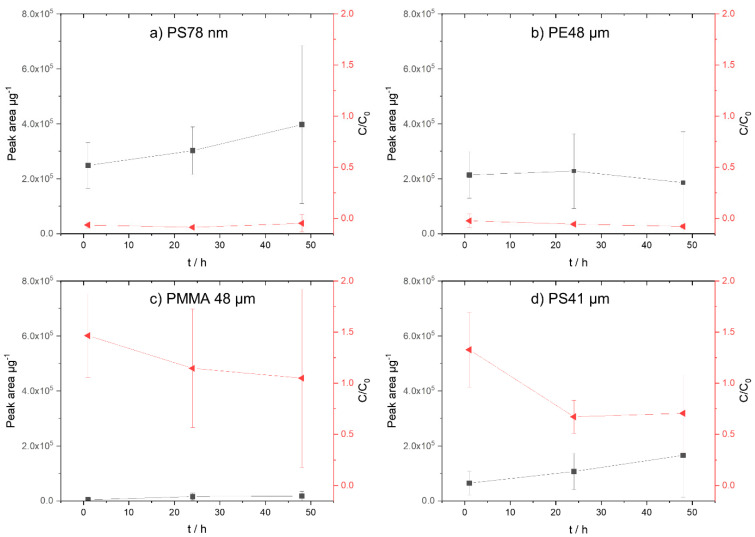
Concentration of phenanthrene on micro- and nanoplastic particles. On the left *y*−axis (black, square), the sorption on the particles (peak area µg^−1^) is shown; on the right *y*−axis (red, triangle), the normalized concentration in the aqueous phase (C/C_0_) is presented. (**a**) PS 78 nm, (**b**) PE 48 µm, (**c**) PMMA 48 µm, and (**d**) PS41 µm.

**Figure 3 biomolecules-12-00953-f003:**
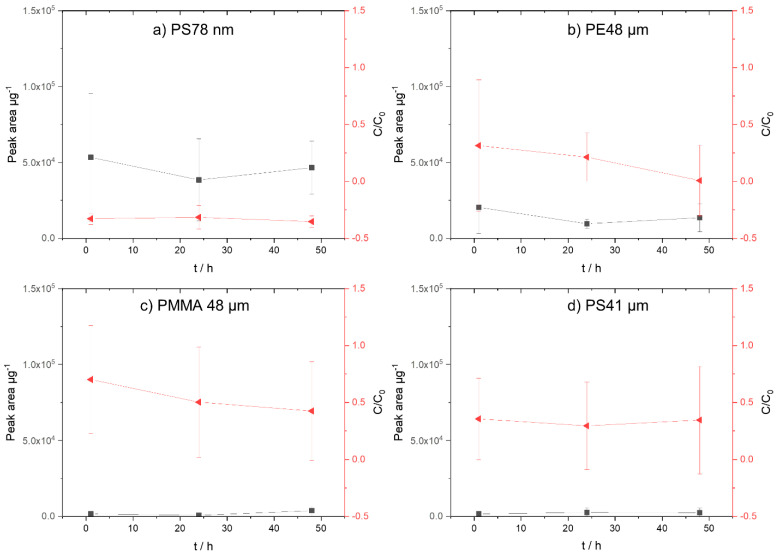
Concentration of triclosan on micro- and nanoplastic particles. On the left *y*−axis (black, square), the sorption on the particles (peak area µg^−1^) is shown; on the right y−axis (red, triangle) the concentration in the aqueous phase (C/C_0_) is presented. (**a**) PS 78 nm, (**b**) PE 48 µm, (**c**) PMMA 48 µm, and (**d**) PS41 µm.

**Figure 4 biomolecules-12-00953-f004:**
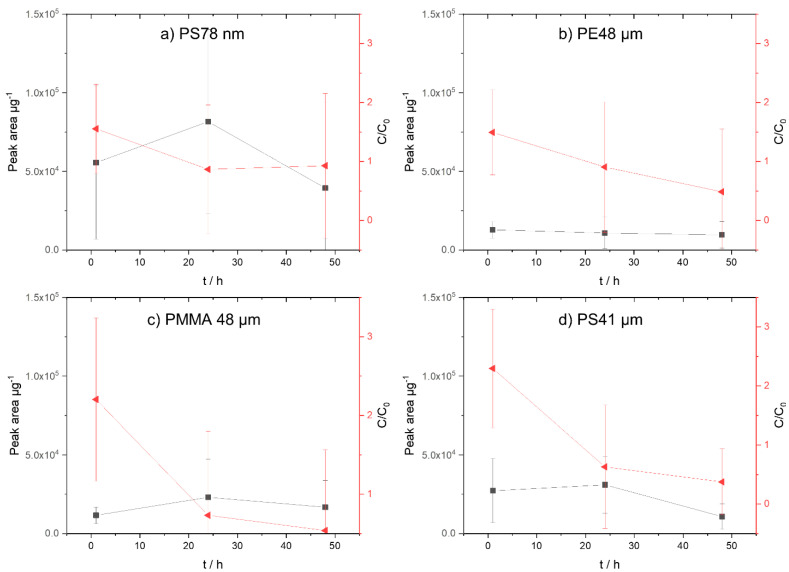
Concentration of α-cypermethrin on micro- and nanoplastic particles. On the left *y*−axis (black, square), the sorption on the particles (peak area µg^−1^) is shown; on the right *y*−axis (red, triangle), the concentration in the aqueous phase (C/C_0_) is presented. (**a**) PS 78 nm, (**b**) PE 48 µm, (**c**) PMMA 48 µm, and (**d**) PS41 µm.

**Figure 5 biomolecules-12-00953-f005:**
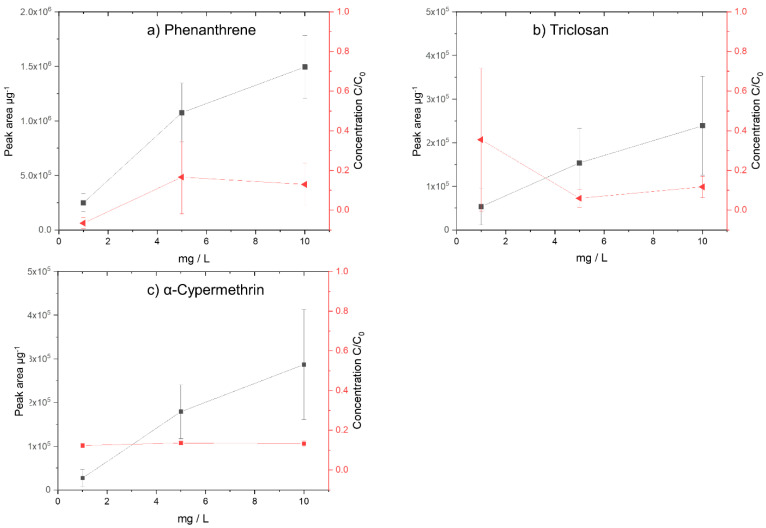
Analysis of initial concentration of 1 mg/L, 5 mg/L, and 10 mg/L in the aqueous phases via TD-GC/MS (C/C_0_) and the particle phases via TD-Pyr-GC/MS (peak area µg^−1^) for the TOrCs (**a**) phenanthrene, (**b**) triclosan, and (**c**) α-cypermethrin. On the left *y*-axis (black, square), the sorption on the particles (peak area µg^−^^1^) is shown; on the right *y*-axis (red, triangle) the concentration in the aqueous phase (C/C_0_) is presented.

**Figure 6 biomolecules-12-00953-f006:**
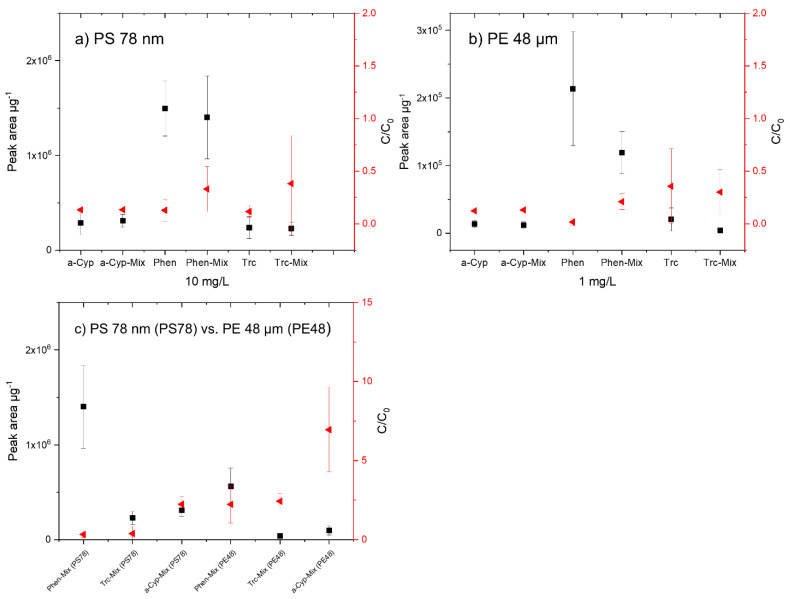
Comparison of the results of the measured single substances (α-cypermethrin (a-Cyp), Phenanthrene (Phen), and Triclosan (Trc)) and the measurement of the mixed TOrCS (a-Cyp-Mix, Phen-Mix, Trc-Mix) with an initial concentration of 10 mg/L and sorption on PS 78 nm (**a**), with an initial concentration of 1 mg/L and sorption on PE 48 µm particles (**b**), and with an initial concentration of 10 mg/L on PS78 nm and PE 48 µm (**c**); measurements of the aqueous phase were conducted via Gerstel Twister^®^ and TD-GC/MS measurements of the particles were carried out with TD-Pyr-GC/MS. On the left *y*−axis (black, square), the sorption on the particles (peak area µg^−1^) is shown; on the right *y*−axis (red, triangle), the concentration in the aqueous phase (C/C_0_) is presented.

**Table 1 biomolecules-12-00953-t001:** Characteristic signals for MS analysis, properties, and structure of selected TOrCs.

Substance	Characteristic Signals (*m*/*z*)	Molecular Weight (g/mol)	Van der Waals Surface * (Å^2^)	log D (pH 7) *	Structure
α-Cypermethrin	163, 184, 209	416	571	5.35	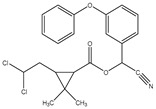
Phenanthrene	178	178	261	3.95	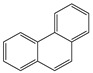
Triclosan	290, 288, 218, 63	290	319	5.80	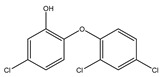

* calculated via MarvinSketch 21.3. All data were statistically analyzed for significant outliers using Dixon’s Q-test and Grubbs test at a level of 0.05. The significant outliers were no longer included in data evaluation.

**Table 2 biomolecules-12-00953-t002:** Ratio between TOrC surface area and particle surface area for TOrCs (phenanthrene, triclosan, α-cypermethrin) amount of 10 µg (=1 mg/L) each and a particle amount (PS78 nm (PS78), PS 41 µm (PS41), PE 48 µm (PE), and PMMA 48 µm (PMMA)) of 10 mg.

	Particle	PS 78 nm Surface: 0.740 m^2^	PS 41 µm Surface: 0.0014 m^2^	PE 48 µm Surface: 0.0014 m^2^	PMMA 48 µm Surface: 0.0011 m^2^
TOrC					
		Ratio TOrC/particle			
**Phenanthrene** **Surface: 0.09 m^2^**		0.12	62.1	64.8	83.2
**Triclosan** **Surface: 0.0056 m^2^**		0.07	38.6	39.9	51.2
**α-Cypermethrin Surface: 0.0039 m^2^**		0.05	26.8	27.8	35.6

## Data Availability

Data is contained within the article.

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
