# Peer review of "A Novel Analytical Approach to Assessing Sorption of Trace Organic Compounds into Micro- and Nanoplastic Particles"

_biomolecules, 2022, doi:10.3390/biom12070953_

Round 1
Reviewer 1 Report
The paper was interesting to read and I suggest that it is published following very minor revisions:
Introduction - A number of persistent organic pollutants HAVE been found
Improve the resolution of Figure 1
Explain the different coloured plots more clearly in captions.
Author Response
Reviewer 1:
- Introduction – A number of persistent organic pollutants HAVE been found:
Response: The sentence was corrected.
- Improve the resolution of Figure 1:
Response: The resolution of Figure 1 has been increased.
- Explain the different coloured plots more clearly in captions:
Response: In the caption, the plots were described in more detail by indicating the color and shape of the data points, e.g. "black, square".
Reviewer 2 Report
Dear Authors,
I my opinion this article very interesting and helpful. The scope of the present work was aims to presented combined concept of TD-GC/MS and TD-Pyr-GC/MS analysis. In all experiments, both the aqueous phase (TD-GC/MS) and the particles (TD-Pyr-GC/MS) were analyzed directly. The aim of this study was to investigate the degree of sorption of each TOrCs after 1 h, 24 h, and 48 h on representative micro- and nanoparticles. The manuscript is well organized. The problem statements agree with the title and have significance. The methods used to gather the data for this article were clearly explained. Overall, it was a very interesting, significant contribution to the field of research. The quality of citations is good. However, some aspects must be revised before the acceptation for this Journal.
Some specific comments are listed below.
1. Please check marking of polymeric materials because in my opinion there are inconsistencies (PE, PMMA, whether such a marking is to apply PS41?, PS78?)
2. In figures 2, 3, 4, 6, check markings.
3. Charts and diagrams need correction - see Figures (esspecially Figure 6 c) are invisible and thus difficult to interpret
Please check the numbering of the figures, tables because it doesn't match.
4. Why in some places units are written with a space and in anohther place without space?
Author Response
Reviewer 2:
- Please check marking of polymeric materials because in my opinion there are inconsistencies (PE, PMMA, whether such marking is to apply PS41?, PS78?)
Response: The marking of the polymers was standardized as: PMMA 48 µm, PE 48 µm, PS 41 µm and PS 78 nm.
- In figures 2,3,4,6, check markings
Response: In all figures, the data points were described in more specific color and shape, e.g. "black, square", to ensure better labeling.
- Charts and diagrams need correction – see Figures (especially Figure 6c) are invisible and thus difficult to interpret. Please check the numbering of the figures, tables because it doesn’t match
Response: The resolution of all figures was increased to provide a better readability of the data. In addition, the numbering of the figures and tables was checked again and adjusted if necessary (e.g. table 3 corrected in table 2).
- Why in some places units are written with a space and in another place without space?
- Response: All units have been unified. Also the spelling of the specific particle descriptions, e.g. "PS 78 nm".
Reviewer 3 Report
The aim of the study was to establish the analysis of TOrCs directly from particles and validate it with an aqueous phase analysis. The manuscript is well written.
Page 14: "By means of TD-Pyr-C/MS, quantitative experiments especially regarding .." GC?
Author Response
Reviewer 3:
- Page 14: “By means of TD-Pyr-C/MS, quantitative experiments especially regarding…” GC?
Response: The typo has been corrected.